# IMPLICIT NEURAL REPRESENTATIONS AND THE ALGEBRA OF COMPLEX WAVELETS

**T. Mitchell Roddenberry, Vishwanath Saragadam,**[*]
**Maarten V. de Hoop, Richard G. Baraniuk**
Rice University
Houston, TX, USA
{mitch,mvd2,richb}@rice.edu, vishwanath.saragadam@ucr.edu

## ABSTRACT

Implicit neural representations (INRs) have arisen as useful methods for representing signals on Euclidean domains. By parameterizing an image as a multi-layer perceptron (MLP) on Euclidean space, INRs effectively couple spatial and spectral features of the represented signal in a way that is not obvious in the usual discrete representation. Although INRs using sinusoidal activation functions have been studied in terms of Fourier theory, recent works have shown the advantage of using wavelets instead of sinusoids as activation functions, due to their ability to simultaneously localize in both frequency and space. In this work, we approach such INRs and demonstrate how they resolve high-frequency features of signals from coarse approximations performed in the first layer of the MLP. This leads to multiple prescriptions for the design of INR architectures, including the use of progressive wavelets, decoupling of low and high-pass approximations, and initialization schemes based on the singularities of the target signal.

## 1 INTRODUCTION

Implicit neural representations (INRs) are a powerful set of neural architectures for representing and processing signals on low-dimensional spaces. By learning a continuous interpolant of a set of sampled points, INRs have enabled and advanced state-of-the-art methods in signal processing (Xu et al., 2022) and computer vision (Mildenhall et al., 2020).

Typical INRs are specially designed multilayer perceptrons (MLPs), where the activation functions are chosen in such a way to yield a desirable signal representation. Although INRs often can be easily understood at the first layer due to the simplicity of plotting the function associated to each neuron based on its weights and biases, the behavior of the network in the second layer and beyond is more opaque, apart from some theoretical developments in the particular case of a sinusoidal first layer (Yüce et al., 2022).

This work develops a broader theoretical understanding of INR architectures with a wider class of activation functions, followed by practical prescriptions rooted in time-frequency analysis. In particular, we

1. Characterize the function class of INRs in terms of Fourier convolutions of the neurons in the first layer (Lemma 1)

2. Demonstrate how INRs that use complex wavelet functions preserve useful properties of the wavelet, even after the application of the nonlinearities (Corollary 4)

3. Suggest a split architecture for approximating signals that decouples the smooth and nonsmooth parts into linear and nonlinear INRs, respectively (Section 4.3)

4. Leverage connections with wavelet theory to propose efficient initialization schemes for wavelet INRs based on the wavelet modulus maxima for capturing singularities in the target function (Section 5).

---

[*]Now affiliated with UC Riverside.

Following a brief survey of INR methods, the class of architectures we study is defined in Section 2. The main result bounding the function class represented by these architectures is stated in Section 3, which is then related to the algebra of complex wavelets in Section 4. The use of the wavelet modulus maxima for initialization of wavelet INRs is described and demonstrated in Section 5, before concluding in Section 6.

## 2  IMPLICIT NEURAL REPRESENTATIONS

Wavelets as activation functions in MLPs have been shown to yield good function approximators (Zhang & Benveniste, 1992; Marar et al., 1996). These works have leveraged the sparse representation of functions by wavelet dictionaries in order to construct simple neural architectures and training algorithms for effective signal representation. Indeed, an approximation of a signal by a finite linear combination of ridgelets (Candès, 1998) can be viewed as one such MLP using wavelet activation functions. Additionally, wavelets have been used to study the expressivity of deep neural networks and their approximation capacity for functions on manifolds (Shaham et al., 2018), for instance.

Recently, sinusoidal activation functions in the first layer (Tancik et al., 2020) and beyond (Sitzmann et al., 2020; Fathony et al., 2020) have been shown to yield good function approximators, coupled with a harmonic analysis-type bound on the function class represented by these networks (Yüce et al., 2022). Similar to the Fourier embedding of the coordinate space that is done by methods such as SIREN (Sitzmann et al., 2020), eigenvectors of graph operators have been used to define INRs for signals on more general spaces (Grattarola & Vandergheynst, 2022). Other methods have used activation functions that, unlike sinusoids, are localized in space, such as gaussians (Ramasinghe & Lucey, 2021) or Gabor wavelets (Saragadam et al., 2023).

Following the formulation of (Yüce et al., 2022), we define an INR to be a map $f_{\boldsymbol{\theta}} : \mathbb{R}^d \to \mathbb{C}$ defined in terms of a function $\psi : \mathbb{R}^d \to \mathbb{C}$, followed by an MLP with analytic[1] activation functions $\rho^{(\ell)} : \mathbb{C} \to \mathbb{C}$ for layers $\ell = 1, \ldots, L$:

$$
\begin{aligned}
\mathbf{z}^{(0)}(\mathbf{r}) &= \psi(\mathbf{W}^{(0)}\mathbf{r} + \mathbf{b}^{(0)}) \\
\mathbf{z}^{(\ell)}(\mathbf{r}) &= \rho^{(\ell)}(\mathbf{W}^{(\ell)}\mathbf{z}^{(\ell-1)}(\mathbf{r}) + \mathbf{b}^{(\ell)}) \\
f_{\boldsymbol{\theta}}(\mathbf{r}) &= \mathbf{W}^{(L)}\mathbf{z}^{(L-1)}(\mathbf{r}) + \mathbf{b}^{(L)},
\end{aligned}
\tag{1}
$$

where $\boldsymbol{\theta}$ denotes the set of parameters dictating the tensor $\mathbf{W}^{(0)} \in \mathbb{R}^{F_1 \times d \times d}$, matrices $\mathbf{W}^{(\ell)} \in \mathbb{C}^{F_{\ell+1} \times F_\ell}$ and $\mathbf{b}^{(0)} \in \mathbb{R}^{F_1 \times d}$, and vectors $\mathbf{b}^{(\ell)} \in \mathbb{C}^{F_{\ell+1}}$ for $\ell = 1, \ldots, L$, with fixed integers $F_\ell$ satisfying $F_{L+1} = 1$. The function $\psi : \mathbb{R}^d \to \mathbb{C}$ is understood to act on $\mathbf{W}^{(0)}\mathbf{r} + \mathbf{b}^{(0)}$ row-wise, *i.e.*, as a map $\psi : \mathbb{R}^{F_1 \times d} \to \mathbb{C}^{F_1}$. We will henceforth refer to $\psi$ as the *template function* of the INR. Owing to the use of Gabor wavelets by Saragadam et al. (2023), we will refer to functions of the form (1) as *WIRE INRs*, although (1) also captures architectures that do not use wavelets, such as SIREN (Sitzmann et al., 2020).

## 3  EXPRESSIVITY OF INRs

For the application of INRs to practical problems, it is important to understand the function class that an INR architecture can represent. We will demonstrate how the function parameterized by an INR can be understood via time-frequency analysis, ultimately motivating the use of wavelets as template functions.

Noting that polynomials of sinusoids generate linear combinations of integer harmonics of said sinusoids, Yüce et al. (2022) bounded the expressivity of SIREN (Sitzmann et al., 2020) and related architectures (Fathony et al., 2020). These results essentially followed from identities relating products of trigonometric functions. For template functions that are not sinusoids, such as wavelets (Saragadam et al., 2023), these identities do not hold. The following result offers a bound on the class of functions represented by an INR.

---

[1]That is, entire on $\mathbb{C}$.

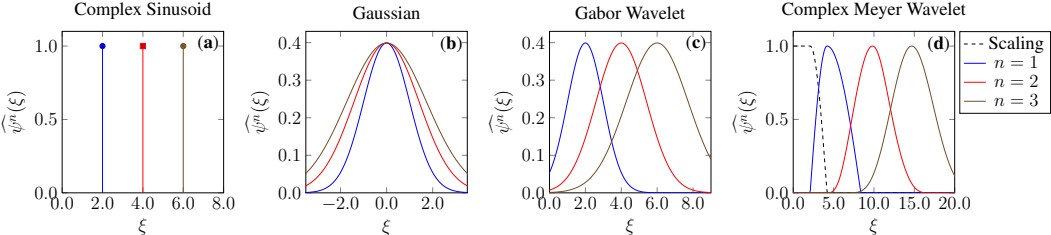

Figure 1: Fourier transforms of template functions $\psi$ and their powers $\psi^n$.

**Lemma 1.** *Let $f_{\boldsymbol{\theta}} : \mathbb{R}^d \to \mathbb{C}$ be a WIRE INR. Assume that each of the activation functions $\rho^{(\ell)}$ is a polynomial of degree at most $K$, and that the Fourier transform of the template function $\psi$ exists.*[2] *Let $\mathbf{W}^{(0)}\mathbf{r} = [\mathbf{W}_1\mathbf{r}, \ldots, \mathbf{W}_{F_1}\mathbf{r}]^\top$ for $\mathbf{W}_1, \ldots, \mathbf{W}_{F_1} \in \mathbb{R}^{d \times d}$ each having full rank, and also let $\mathbf{b}^{(0)} = [\mathbf{b}_1, \ldots, \mathbf{b}_{F_1}]^\top$ for $\mathbf{b}_1, \ldots, \mathbf{b}_{F_1} \in \mathbb{R}^d$. For $k \geq 0$, denote by $\Delta(F_1, k)$ the set of ordered $F_1$-tuples of nonnegative integers $\boldsymbol{m} = [m_1, \ldots, m_{F_1}]$ such that $\sum_{t=1}^{F_1} m_t = k$.*

*Let a point $\mathbf{r}_0 \in \mathbb{R}^d$ be given. Then, there exists an open neighborhood $U \ni \mathbf{r}_0$ such that for all $\phi \in \mathcal{C}_0^\infty(U)$*

$$\widehat{\phi \cdot f_{\boldsymbol{\theta}}}(\xi) = \left( \widehat{\phi} * \sum_{k=0}^{K^{L-1}} \sum_{\boldsymbol{m} \in \Delta(F_1, k)} \widehat{\beta}_{\boldsymbol{m}} \underset{t=1}{\overset{F_1}{\boldsymbol{\bigast}}} \left( e^{i2\pi\langle \mathbf{W}_t^{-\top}\xi, \mathbf{b}_t \rangle} \widehat{\psi}(\mathbf{W}_t^{-\top}\xi) \right)^{*m_t, \xi} \right)(\xi), \qquad (2)$$

*for coefficients $\widehat{\beta}_{\boldsymbol{m}} \in \mathbb{C}$ independent of the choice of $\mathbf{r}_0 \in U$, where $(\cdot)^{*m, \xi}$ denotes $m$-fold convolution[3] of the argument with itself with respect to $\xi$, and $\mathcal{C}_0^\infty(U)$ denotes the set of all infinitely differentiable functions with compact support contained in $U$. Furthermore, the coefficients $\widehat{\beta}_{\boldsymbol{m}}$ are only nonzero when each $t \in [1, \ldots, F_1]$ such that $m_t \neq 0$ also satisfies $\mathbf{W}_t \mathbf{r}_0 + \mathbf{b}_t \in \text{supp}(\psi)$.*

The proof, a simple application of the convolution theorem, is left to Appendix A. Lemma 1 illustrates two things. First, the output of an INR has a Fourier transform determined by convolutions of the Fourier transforms of the atoms in the first layer with themselves, serving to generate "integer harmonics" of the initial atoms determined by scaled, shifted copies of the template function $\psi$. Notably, this recovers (Yüce et al., 2022, Theorem 1). Second, the support of these scaled and shifted atoms is preserved, so that the output at a given coordinate $\mathbf{r}$ is dependent only upon the atoms in the first layer whose support contains $\mathbf{r}$.

*Remark* 2. The assumptions behind Lemma 1 can be relaxed to capture a broader class of architectures. By imposing continuity conditions on the template function $\psi$, the activation functions can be reasonably extended to analytic functions. These extensions are discussed in Appendix B.

## 4  THE ALGEBRA OF COMPLEX WAVELETS

Of the INR architectures surveyed in Section 2, the only one to use a complex wavelet template function is WIRE (Saragadam et al., 2023), where a Gabor wavelet is used. Gabor wavelets are essentially band-pass filters and are necessarily complex-valued due to their lack of conjugate symmetry in the Fourier domain. We now consider the advantages of using complex wavelets, or more precisely *progressive wavelets*, as template functions for INRs by examining their structure as an algebra of functions.

---

[2]Even if only in the sense of tempered distributions.

[3]0-fold convolution is defined by convention to yield the Dirac delta. We also use the symbol $\boldsymbol{\bigast}$ to denote the convolution of several functions, in this case indexed by $t = 1, \ldots, F_1$.

## 4.1 PROGRESSIVE TEMPLATE FUNCTIONS

For the sake of discussion, suppose that $d = 1$, so that the INR represents a 1D function. The template function $\psi : \mathbb{R} \to \mathbb{C}$ is said to be *progressive*[4] if it has no negative frequency components, *i.e.*, for $\xi < 0$, we have $\widehat{\psi}(\xi) = 0$ (Mallat, 1999). No nonzero real-valued function is progressive. It is obvious that progressive functions remain progressive under scalar multiplication, shifts, and positive scaling. That is, for arbitrary $s > 0, u \in \mathbb{R}, z \in \mathbb{C}$, if $\psi(x)$ is progressive, then the function $z \cdot D_s T_u \psi(x) := z \cdot \psi((x - u)/s)$ is also progressive. Moreover, progressive functions are closed under multiplication, so that if $\psi_1$ and $\psi_2$ are progressive, then $\psi_3(x) := \psi_1(x)\psi_2(x)$ is also progressive,[5] *i.e.*, progressive functions constitute an algebra over $\mathbb{C}$.

**Example 1** (Complex Sinusoid). For any $\omega > 0$, the complex sinusoid $\psi(x; \omega) = \exp(-i2\pi\omega x)$ is a progressive function, as its Fourier transform is a Dirac delta centered at $\omega$. As pictured in Fig. 1 (a), the exponents $\psi^n(\cdot; \omega)$ are themselves complex sinusoids, where $\psi^n(x; \omega) = \psi(x; n\omega)$.

**Example 2** (Gaussian). The gaussian function, defined for some $\sigma > 0$ as $\psi(x; \sigma) = \exp(-x^2/(2\sigma^2))$, is *not* a progressive function, as its Fourier transform is symmetric and centered about zero. Moreover, as pictured in Fig. 1 (b), the exponents are also gaussian functions $\psi^n(x; \sigma) = \psi(x; \sigma/\sqrt{n})$, which also have Fourier transform centered at zero. Unlike the complex sinusoid, the powers of the gaussian are all low-pass, but with increasingly wide passband.

**Example 3** (Gabor Wavelet). For any $\omega, \sigma > 0$, the Gabor wavelet defined as $\psi(x; \omega, \sigma) = \exp(-x^2/(2\sigma^2) - i2\pi\omega x)$ is *not* a progressive function, as its Fourier transform is a gaussian centered at $\omega$ with standard deviation $1/\sigma$. However, the Fourier transform of the exponents $\psi^n$ for integers $n > 0$ are gaussians centered at $n\omega$ with standard deviation $\sqrt{n}/\sigma$, as pictured in Fig. 1 (c). So, as $n$ grows sufficiently large, the effective support of $\widehat{\psi^n}$ will be contained in the positive reals, so that the Gabor wavelet can be considered as a progressive function for the purposes of studying INRs.

A progressive function on $\mathbb{R}$ has Fourier support contained in the nonnegative real numbers. Of course, there is not an obvious notion of nonnegativity that generalizes to $\mathbb{R}^d$ for $d > 1$. Noting that the nonnegative reals form a convex conic subset of $\mathbb{R}$, we define the notion of a progressive function with respect to some conic subset of $\mathbb{R}^d$:

**Definition 3.** Let $\Gamma \subseteq \widehat{\mathbb{R}}^d$ be a convex conic set, *i.e.*, for all $\gamma_1, \gamma_2 \in \Gamma$ and $a_1, a_2 \geq 0$, we have that $a_1\gamma_1 + a_2\gamma_2 \in \Gamma$.[6] A function $\psi : \mathbb{R}^d \to \mathbb{C}$ is said to be $\Gamma$-*progressive* if $\mathrm{supp}(\widehat{\psi}) \subseteq \Gamma$. The function $\psi$ is said to be *locally $\Gamma$-progressive at* $\mathbf{r}_0 \in \mathbb{R}^d$ if there exists some $\Gamma$-progressive function $\psi_{\mathbf{r}_0} : \mathbb{R}^d \to \mathbb{C}$ so that for all smooth functions $\phi \in \mathcal{C}_0^\infty(\mathbb{R}^d)$ with support in a sufficiently small neighborhood of $\mathbf{r}$, we have

$$\widehat{\phi \cdot \psi} = \widehat{\phi} * \widehat{\psi_{\mathbf{r}_0}}. \tag{3}$$

Curvelets (Candès & Donoho, 2004), for instance, are typically defined in a way to make them $\Gamma$-progressive for some conic set $\Gamma$ that indicates the oscillatory direction of a curvelet atom. Observe that if $\Gamma$ is a conic set, then for any matrix $\mathbf{W}$, the set $\mathbf{W}^\top \Gamma$ is also conic. Thus, for a function $\psi$ that is $\Gamma$-progressive, the function $\psi(\mathbf{W}x)$ is $\mathbf{W}^\top \Gamma$-progressive.

Observe further that for two $\Gamma$-progressive functions $\psi_1, \psi_2$, their product is also $\Gamma$-progressive. The closure of progressive functions under multiplication implies that an analytic function applied pointwise to a progressive function is progressive. For INRs as defined in (1), this yields the following corollary to Lemma 1.

**Corollary 4.** *Let $\Gamma \subseteq \widehat{\mathbb{R}}^d$ be conic, and let $\psi : \mathbb{R}^d \to \mathbb{C}$ be given, with Fourier support denoted $\Gamma_0 := \mathrm{supp}(\widehat{\psi})$. Let $\mathbf{W}^{(0)}\mathbf{r} = [\mathbf{W}_1\mathbf{r}, \dots, \mathbf{W}_{F_1}\mathbf{r}]^\top$ for $\mathbf{W}_1, \dots, \mathbf{W}_{F_1} \in \mathbb{R}^{d \times d}$ each having full rank. Assume that for each $t = 1, \dots, F_1$, we have $\mathbf{W}_t^\top \Gamma_0 \subseteq \Gamma$. Then, the WIRE INR $f_\theta : \mathbb{R}^d \to \mathbb{C}$ defined by (1) is a $\Gamma$-progressive function.*

*Moreover, if we fix some $\mathbf{r}_0 \in \mathbb{R}^d$, and if the assumption $\mathbf{W}_t^\top \Gamma_0 \subseteq \Gamma$ holds for the indices $t$ such that $\mathbf{W}_t\mathbf{r} + \mathbf{b}_t \in \mathrm{supp}(\psi)$, then $f_\theta$ is locally $\Gamma$-progressive at $\mathbf{r}_0$.*

---

[4]More commonly known as an *analytic signal*, we use this terminology (following Grossmann et al. (1990)) to avoid confusion with the analytic activation functions used in the INR.

[5]This is a simple consequence of the convolution theorem.

[6]We henceforth refer to such sets as simply "conic."

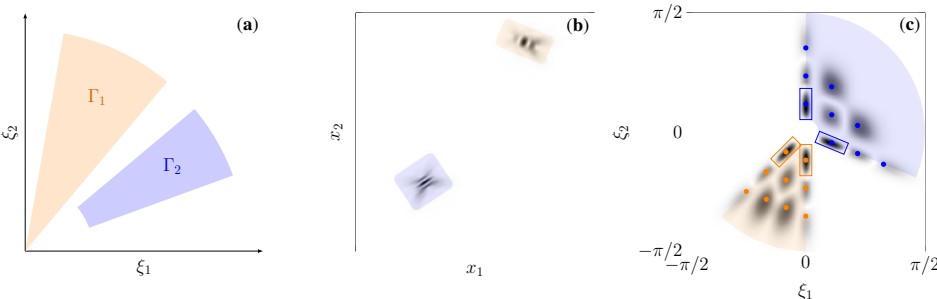

Figure 2: **(a)** A conic set $\Gamma_1 \subset \mathbb{R}^2$, and a weakly conic set $\Gamma_2 \subset \mathbb{R}^2$ (both truncated for illustration purposes). **(b)** Modulus of a function $g$ given by the sum of four template atoms. **(c)** Fourier transform of $f_{\boldsymbol{\theta}}(\mathbf{r}) = \rho(g(\mathbf{r}))$, where $\rho(z) = -z + z^2 - z^3$. The blue and orange cones correspond to the respectively highlighted parts of the function $g$. Effective Fourier supports of the template atoms constituting $g$ are enclosed by rectangles, and approximate centers of Frequency support for each atom and product of atoms are marked by colored circles.

The proof is left to Appendix C; essentially, the property of $\Gamma$-progressive functions constituting an algebra over $\mathbb{C}$ combined with the polynomial structure of the INR is shown to preserve the $\Gamma$-progressive property. Thus, any advantages/limitations of approximating functions using $\Gamma$-progressive template functions are maintained.

*Remark* 5. One may notice that a $\Gamma$-progressive function will always incur a large error when approximating a real-valued function, as real-valued functions have conjugate symmetric Fourier transforms (apart from the case $\Gamma = \widehat{\mathbb{R}}^d$). For fitting real-valued functions, it is effective to simply fit the *real part* of the INR output to the function, as taking the real part of a function symmetrizes it in the Fourier domain. In the particular case of $d = 1$, fitting the real part of a progressive INR to a function is equivalent to fitting the INR to that function's Hilbert transform.

## 4.2 BAND-PASS PROGRESSIVE WAVELETS

Corollary 4 holds for conic sets $\Gamma$, but is also true for a larger class of sets. If some set $\Gamma \subseteq \mathbb{R}^d$ is conic, it is by definition closed under all sums with nonnegative coefficients. Alternatively, consider the following weaker property:

**Definition 6.** Let $\Gamma \subseteq \widehat{\mathbb{R}}^d$. $\Gamma$ is said to be *weakly conic* if for all $\gamma_1, \gamma_2 \in \Gamma$ and $a_1, a_2 \geq 1$, we have that $a_1\gamma_1 + a_2\gamma_2 \in \Gamma$, and that $0 \in \Gamma$. A function $\psi : \mathbb{R}^d \to \mathbb{C}$ is said to be $\Gamma$-*progressive* if $\text{supp}(\widehat{\psi}) \subseteq \Gamma$. The function $\psi$ is said to be *locally* $\Gamma$-*progressive at* $\mathbf{r}_0 \in \mathbb{R}^d$ if there exists some $\Gamma$-progressive function $\psi_{\mathbf{r}_0} : \mathbb{R}^d \to \mathbb{C}$ so that for all smooth functions $\phi \in \mathcal{C}^\infty(\mathbb{R}^d)$ with support in a sufficiently small[7] neighborhood of $\mathbf{r}$, we have

$$\widehat{\phi \cdot \psi} = \widehat{\phi} * \widehat{\psi_{\mathbf{r}_0}}. \tag{4}$$

The notion of a weakly conic set is illustrated in Fig. 2 (a). Just as in the case of progressive functions for a conic set, the set of $\Gamma$-progressive functions for a weakly conic set $\Gamma \subseteq \widehat{\mathbb{R}}^d$ constitutes an algebra over $\mathbb{C}$. One can check, then, that Corollary 4 holds for weakly conic sets as well. Putting this into context, consider a template function $\psi$ such that $\widehat{\psi}$ vanishes in some neighborhood of the origin. Assume furthermore that $\text{supp}(\widehat{\psi})$ is contained in some weakly conic set $\Gamma$.

**Example 4** (Complex Meyer Wavelet)**.** The complex Meyer wavelet is most easily defined in terms of its Fourier transform. Define

$$\widehat{\psi}(\xi) := \begin{cases} \sin(\frac{3\xi}{4} - \pi/2) & \xi \in [2\pi/3, 4\pi/3] \\ \cos(\frac{3\xi}{8} - \pi/2) & \xi \in [4\pi/3, 8\pi/3] \\ 0 & \text{otherwise.} \end{cases}$$

---

[7]Again, not necessarily compact.

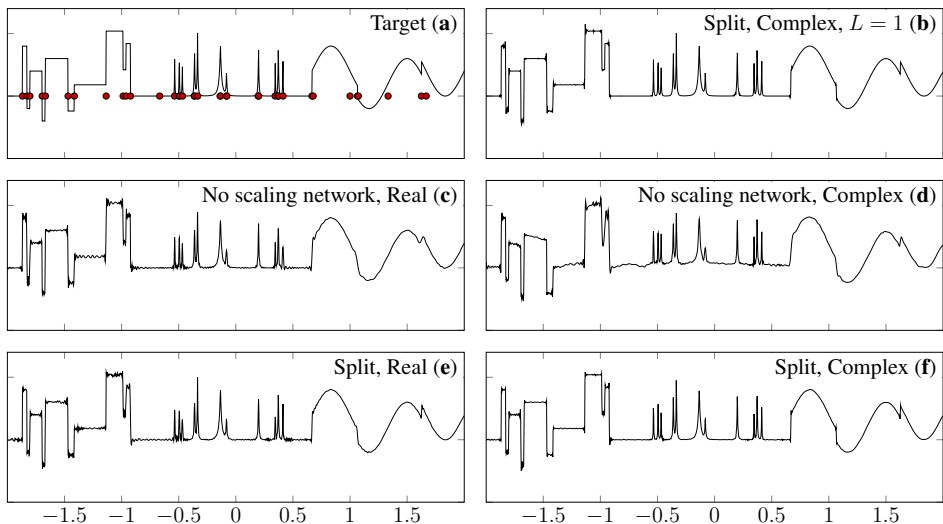

Figure 3: **(a)** Target signal, sampled at $n = 512$ uniformly spaced points in the interval $[-2, 2]$. Wavelet modulus maxima are marked in red. **(b)** Split complex wavelet INR with no hidden layers (MSE = 0.0016). **(c)** "Real" wavelet INR with no scaling network (MSE = 0.0096). **(d)** Complex wavelet INR with no scaling network (MSE = 0.0606). **(e)** Split "real" wavelet INR (MSE = 0.0061). **(f)** Split complex wavelet INR (MSE = **0.0011**).

The complex Meyer wavelet and its exponents are pictured in Fig. 1 (d). Observe that these functions are not only progressive, but are also $\Gamma$-progressive for the weakly conic set $\Gamma = [2\pi/3, \infty)$. The Meyer scaling function, pictured by the dashed line in Fig. 1 (d), has Fourier support that only overlaps that of the complex Meyer wavelet, but none of its powers.

Applying this extension of Corollary 4, we see that if the atoms in the first layer of an INR using such a function $\psi$ have vanishing Fourier transform in some neighborhood of the origin, then the output of the INR has Fourier support that also vanishes in that neighborhood. We illustrate this in $\mathbb{R}^2$ using a template function $\psi : \mathbb{R}^2 \rightarrow \mathbb{C}$ where $\psi$ is the tensor product of a gaussian and a complex Meyer wavelet. Using this template function, we construct an INR with $F_1 = 4$ in the first layer, and a single polynomial activation function. The modulus of the sum of the template functions before applying the activation function is shown in Fig. 2 (b). We then plot the modulus of the Fourier transform of $f_{\boldsymbol{\theta}}$ in Fig. 2 (c). First, observe that since the effective supports of the transformed template functions are supported by two disjoint sets, the Fourier transform of $f_{\boldsymbol{\theta}}$ can be separated into two cones, each corresponding to a region in $\mathbb{R}^2$. Second, since the complex Meyer wavelet vanishes in a neighborhood of the origin, these cones are weakly conic, so that the Fourier transform of $f_{\boldsymbol{\theta}}$ vanishes in a neighborhood of the origin as well, by Corollary 4 applied to weakly conic sets.

*Remark* 7. The weakly conic sets pictured in Fig. 2 (c) are only approximation bounds of the true Fourier support of the constituent atoms. We see that Corollary 4 still holds in an approximate sense, as the bulk of the Fourier support of the atoms is contained in each of the pictured cones.

## 4.3 A SPLIT ARCHITECTURE FOR INRS

Based on this property of INRs preserving the band-pass properties of progressive template functions, it is well-motivated to approximate functions using a sum of two INRs: one to handle the low-pass components using a scaling function, and the other to handle the high-pass components using a wavelet. We illustrate this in Fig. 3, where we fit to a classic test signal on $\mathbb{R}$ (Donoho & Johnstone, 1994), pictured in Fig. 3 (a).

The first INR uses a gaussian template function $\psi(x) = \exp(-(\pi x)^2/6)$ with $L = 1$, and the constraint that the weights $\mathbf{W}^{(0)}$ are all equal to one, *i.e.*, the template atoms only vary in their

abscissa. Such a network is essentially a single-layer perceptron (Zhang & Benveniste, 1992) for representing smooth signals. We refer to this network as the "scaling network."

The second INR uses a Gabor template function $\psi(x) = \exp(-(\pi x)^2/6)\exp(-i2\pi x)$ with $L = 3$, where we initialize the weights in the first layer to be positive, thus satisfying the condition of $\mathbf{W}_t^\top \Gamma \subseteq \Gamma$ in Corollary 4 for $\Gamma = \mathbb{R}^+$. Although $\psi$ is not progressive, its Fourier transform has fast decay, so we consider it to be essentially progressive, and thus approximately fulfilling the conditions of Corollary 4. We refer to this network as the "wavelet network," as it is the WIRE architecture (Saragadam et al., 2023) for signals on $\mathbb{R}$.

Denoting the scaling and wavelet networks by $f_{\boldsymbol{\theta},s}, f_{\boldsymbol{\theta},w} : \mathbb{R} \to \mathbb{C}$, respectively, we consider their sum $f_{\boldsymbol{\theta}} = f_{\boldsymbol{\theta},s} + f_{\boldsymbol{\theta},w}$ as a model, and approximate the target signal by taking $\mathrm{Re}\{f_{\boldsymbol{\theta}}\} : \mathbb{R} \to \mathbb{R}$.

The reason for modeling a signal as the sum of a linear scaling INR and a nonlinear INR with a progressive wavelet is apparent in Fig. 1 (d), where the scaling function and powers of a complex Meyer wavelet are pictured. Observe that the portions of the Fourier spectrum covered by the gaussian scaling function and the high powers of the Gabor wavelet (as in an INR, by Lemma 1) are essentially disjoint. To approximate the low-frequency components of a signal using an INR with a progressive wavelet would require large dilations of the atoms in the first layer, in order to force the center frequency of the wavelet towards zero. Moreover, if the progressive wavelet has a Fourier transform that vanishes in a neighborhood of the origin, such dilations will never properly cover a small neighborhood of the origin. This does not arise when using real-valued template functions, as powers of such functions generate low-frequency components. However, this phenomena is not always desirable, as the low-frequency and high-frequency parts of a signal become highly correlated in this regime, a property that is not necessarily true for natural signals.

The idea behind the "split" architecture, then, is to use a simple network to approximate the smooth parts of the target signal, and then a more complicated nonlinear network to approximate the nonsmooth parts of the signal.

We fit an array of INR architectures to the target signal: a split INR architecture with complex Gabor wavelets but no hidden layers ($L = 1$) in the MLP, one with no scaling network and real wavelets, one with no scaling network and complex wavelets, one with a scaling network and real wavelets, and finally our proposed split architecture with complex Gabor wavelets and hidden layers in the MLP. All wavelet networks apart from the first one have two hidden layers ($L = 3$). In the architectures that use real wavelets, we use a gaussian multiplied by a sinusoid, rather than a complex exponential as in the complex Gabor wavelet.

The results of training these architectures are respectively shown in Fig. 3 (b-f). Hyperparameters for each architecture are described in Appendix E.1. We observe slightly better performance, measured in mean squared error (MSE), from the proposed architecture than the split network with no hidden layers, which in turn outperforms the split architecture using real wavelets. This illustrates both the advantages of using complex wavelets over real wavelets, due to their ability to decouple low and high-frequency parts of a signal, as well as the advantage of the hidden layers that couple wavelet coefficients across scales, as stated by Lemma 1. Indeed, the nonlinear activation functions generate "wavelets" from the template function atoms, rather than needing to form a long list of wavelet atoms and their abscissa in a way that does not capture how wavelet coefficients near singularities are correlated across scales.

The two INR architectures without scaling networks fared the worst in this experiment; however, we note that the INR using real wavelets outperformed the one using complex wavelets. This is because powers of real-valued wavelets can generate low-frequency signal content.

To see the role of the nonlinearities in the wavelet INR, we freeze the weights and biases in the first layer of the split complex wavelet INR, and take an optimal linear combination of the resulting template atoms to fit the signal, thus yielding an INR with no hidden layers (Zhang & Benveniste, 1992). We compare the Fourier transforms of the original wavelet network to this "linearized" one in Fig. 4 (b-c), where we see that the nonlinear wavelet network is able to resolve much more high-frequency signal content than the linear one. This reflects how the activation functions resolve high-frequency features from low-frequency approximations, as illustrated initially in Fig. 2. Moreover, the fact that both wavelet networks have the bulk of their Fourier support only on positive frequencies

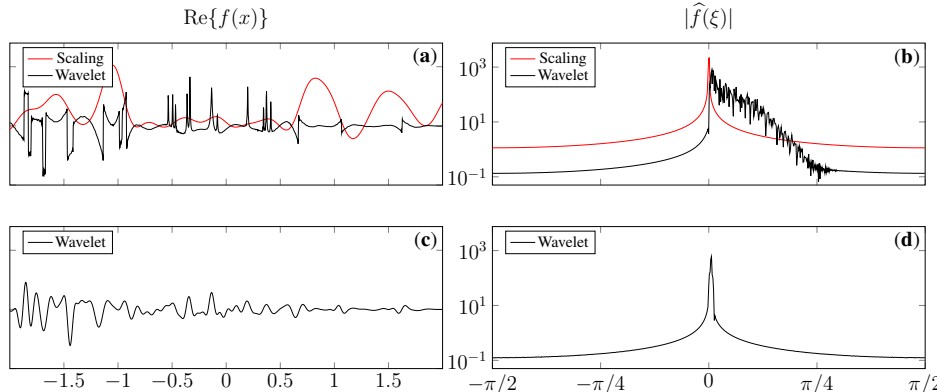

Figure 4: **(a)** Split complex wavelet INR, separated into scaling and wavelet networks. **(b)** Fourier transform of scaling and wavelet networks. **(c)** Linearized complex wavelet INR. **(d)** Fourier transform of linearized wavelet network.

illustrates the how the algebraic closure of progressive wavelets under multiplication applies to INR architectures, as in Corollary 4.

## 5 RESOLUTION OF SINGULARITIES

A useful model for studying sparse representations of images is the *cartoon-like image*, which is a smooth function on $\mathbb{R}^2$ apart from singularities along a twice-differentiable curve (Candès & Donoho, 2004; Wakin et al., 2006).

The smooth part of an image can be handled by the scaling function associated to a wavelet transform, while the singular parts are best captured by the wavelet function. In the context of the proposed split INR architecture, the scaling INR yields a smooth approximation to the signal, and the wavelet INR resolves the remaining singularities. Inspired by this, we now consider how the wavelet INR can be initialized with the resolution of isolated singularities in mind.

### 5.1 INITIALIZATION WITH THE WAVELET MODULUS MAXIMA

As demonstrated by Lemma 1, the function $\psi$ in the first layer of an INR determines the expressivity of the network. Many such networks satisfy a universal approximation property (Zhang & Benveniste, 1992), but their value in practice comes from their implicit bias (Yüce et al., 2022; Saragadam et al., 2023) in representing a particular class of functions. For instance, using a wavelet in the first layer results in sharp resolution of edges with spatially compact error (Saragadam et al., 2023). In the remainder of this section, we demonstrate how an understanding of singular points in terms of the wavelet transform can be used to bolster INR architectures and initialization schemes.

Roughly speaking, isolated singularities in a signal are points where the signal is nonsmooth, but is smooth in a punctured neighborhood around that point. Such singularities generate "wavelet modulus maxima" (WMM) curves in the continuous wavelet transform (Mallat, 1999), which have slow decay in the Fourier domain. With Lemma 1 in mind, we see that INRs can use a collection of low-frequency template atoms and generate a collection of coupled high-frequency atoms, while also preserving the spatial locality of the template atoms.

The combination of these insights suggests a method for the initialization of INRs. In particular, for a given number of template atoms $F_1$ in an INR, the network weights $\mathbf{W}^{(0)}$ and abscissa $\mathbf{b}^{(0)}$ should be initialized in a way that facilitates effective training of the INR via optimization methods. We empirically demonstrate the difference in performance for INRs initialized at random and INRs initialized in accordance with the singularities in the target signal. Once again, we fit the sum of a scaling network and a wavelet network to the target signal in Fig. 3. In Fig. 5 (a), we plot the mean squared error (MSE) for this setting after 1000 training steps for both randomly initialized and strategically initialized INRs, for $F_1 = Km$, where $K \in \{1, 3, 5, 7\}$ and $m$ is the number of

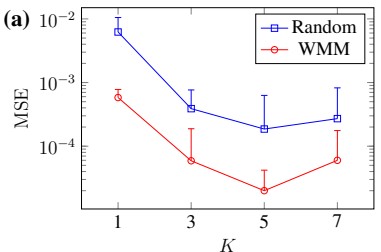

**(a)**

**(b)**

| Setting | PSNR (dB) |
|---|---|
| Gabor, Random Init. (fit) | 30.081 |
| Gabor, WMM Init. (fit) | **30.835** |
| Gaussian (fit) | 26.950 |
| Real Gabor (fit) | 30.826 |
| Sine (fit) | 18.001 |
| Gabor, Random Init. (denoise) | 29.236 |
| Gabor, WMM Init. (denoise) | **29.385** |

Figure 5: **(a)** Mean and standard deviation of the MSE over 10 trials on the 1D test signal. **(b)** PSNR of different architectures and initialization schemes on the Kodak dataset (kod, 1999), for image fitting and denoising tasks. In denoising task, noisy images had average PSNR of 17.35 dB.

WMM points as determined by an estimate of the continuous wavelet transform of the target signal. The randomly initialized INRs have abscissa distributed uniformly at random over the domain of the signal. The strategically initialized INRs place $K$ template atoms at each WMM point (so, a deterministic set of abscissa points). Both initialization schemes randomly distribute the scale weights uniformly in the interval $[1, K]$. We observe that for all $K$, the MSE of the strategically initialized INR is approximately an order of magnitude less than that of the randomly initialized INR.

When $d = 2$, *e.g.*, images, the WMM can be approximated by the gradients of the target signal to obtain an initial set of weights and biases for the wavelet INR. We evaluate this empirically on the Kodak Lossless True Color Image Suite (kod, 1999). We approximate the target images using the proposed split INR architecture. For the WMM-based initialization, we apply a Canny edge detector (Canny, 1986) to encode the positions and directions of the edges. Further details can be found in Appendix D. The architectures used are described in Appendix E.2.

We record the results of this experiment for a variety of template functions and initialization schemes in Fig. 5 (b). Experiments are done for two tasks: image representation, where the network is fit to the ground truth image, and denoising, where the network is fit to a noisy image and the error relative to the ground truth is measured. The latter experiment demonstrates how the implicit bias of INRs is such that they fit natural images better than they fit noise. We observe that over the whole dataset, initializing the network using the WMM yields a higher PSNR for both tasks, demonstrating the utility of smart initialization of INRs. Results are aggregated over the whole dataset; see Appendix F for results on the individual images.

## 6 CONCLUSIONS

We have offered a time-frequency analysis of INRs that leverages polynomial approximations of the nonlinear behavior of MLPs beyond the first layer. By noting that progressive functions form an algebra over the complex numbers, we demonstrated that this analysis yields insights into the behavior of INRs using complex wavelets, such as WIRE (Saragadam et al., 2023). This leads to a split architecture for approximating signals, which decouples the low-pass and high-pass parts of a signal using two INRs, roughly corresponding to the scaling and wavelet functions of a wavelet transform. Furthermore, the connection with the theory of wavelets yields a natural initialization scheme for the weights of an INR based on the singularities of a signal.

INR architectures built using wavelet activation functions offer useful advantages for function approximation that balance locality in space and frequency. The structure of complex wavelets as an algebra of functions with conic Fourier support, combined with the application of INRs for interpolating sampled functions, suggests a connection with *microlocal and semiclassical analysis* (Monard & Stefanov, 2023). This could potentially be understood and improved by incorporating ideas from shearlet and curvelet systems (Labate et al., 2005; Candès & Donoho, 2004). We also foresee the decoupling of the smooth and singular parts of a signal by the split INR architecture having useful properties for solving inverse problems.

## ACKNOWLEDGEMENTS

This work was supported by NSF grants CCF-1911094, IIS-1838177, and IIS-1730574; ONR grants N00014-18-1-2571, N00014-20-1-2534, and MURI N00014-20-1-2787; AFOSR grant FA9550-22-1-0060; and a Vannevar Bush Faculty Fellowship, ONR grant N00014-18-1-2047. Maarten de Hoop gratefully acknowledges support from the Department of Energy under grant DE-SC0020345, the Simons Foundation under the MATH+X program, and the corporate members of the Geo-Mathematical Imaging Group at Rice University.

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

## A  PROOF OF LEMMA 1

Let $\mathbf{r}_0$ be given, and let $I \subseteq \{1, \ldots, F_1\}$ be the set of indices such that $\mathbf{W}_t \mathbf{r}_0 + \mathbf{b}_t \in \text{supp}(\psi)$. Then, there exists an open neighborhood $U \ni \mathbf{r}_0$ such that for all $\mathbf{r} \in U$, $\psi(\mathbf{W}_t \mathbf{r} + \mathbf{b}_t) = 0$ for any $t \notin I$. Thus, for any $\mathbf{r} \in U$, the assumptions on the activation functions imply that $f_{\boldsymbol{\theta}}(\mathbf{r})$ is expressible as a complex multivariate polynomial of $\{\psi(\mathbf{W}_i \mathbf{r} + \mathbf{b}_t)\}_{i \in I}$ with degree at most $K^{L-1}$, *i.e.*,

$$f_{\boldsymbol{\theta}}(\mathbf{r}) = \sum_{k=0}^{K^{L-1}} \sum_{\boldsymbol{\ell} \in \Delta(|I|,k)} \widehat{\beta}_{\boldsymbol{\ell}} \prod_{i \in I} \psi^{\ell_i}(\mathbf{W}_i \mathbf{r} + \mathbf{b}_i).$$

for some set of complex coefficients $\widehat{\beta}_{\boldsymbol{\ell}}$. The desired result follows immediately from the convolution theorem.

## B  RELAXED CONDITIONS ON THE INR ARCHITECTURE

Lemma 1 assumes that the template function $\psi : \mathbb{R}^d \to \mathbb{C}$ has a Fourier transform that exists in the sense of tempered distributions, and that the activation functions are polynomials, which is a stronger condition than merely assuming they are complex analytic. Here, we discuss ways in which this can be relaxed to include more general activation functions, as well as how template functions on Euclidean spaces other than $\mathbb{R}^d$ can be used.

### B.1  RELAXING THE CLASS OF ACTIVATION FUNCTIONS

In Lemma 1, two assumptions are made. The first is that the template function has a Fourier transform that exists, possibly in the sense of tempered distributions. The second is that the activation functions are polynomials of finite degree. However, given that Lemma 1 is a *local* result, mild assumptions on the template function allow for reasonable extension of this result to analytic activation functions.

Indeed, if $\psi$ is assumed to be continuous, then one can take $V \subset U$ to be a compact subset of the neighborhood guaranteed by Lemma 1. It follows, then, that the functions $\{\psi(\mathbf{W}_t \mathbf{r} + \mathbf{b}_t)\}_{t=1}^{F_1}$ are bounded over $V$. Then, if the activation functions are merely assumed to be analytic, then the INR

can be approximated uniformly well over $V$ by finite polynomials. Without repeating the details of the proof, this yields an "infinite-degree" version of Lemma 1, where for any $\phi \in \mathcal{C}_0^\infty(V)$, we have

$$\widehat{\phi \cdot f_{\boldsymbol{\theta}}}(\xi) = \left(\sum_{k=0}^{\infty} \sum_{\boldsymbol{\ell} \in \Delta(F_1, k)} \widehat{\beta}_{\boldsymbol{\ell}} \left[\widehat{\phi} * \mathop{\text{\Large ✳}}_{t=1}^{F_1} \left(e^{i2\pi\langle \mathbf{W}_t^{-\top}\xi, \mathbf{b}_t\rangle}\widehat{\psi}(\mathbf{W}_t^{-\top}\xi)\right)^{*\ell_t, \xi}\right]\right)(\xi).$$

This condition is not strong enough to handle general continuous activation functions on $\mathbb{C}$, since the Stone-Weierstrass theorem for approximating complex continuous functions requires polynomials terms to include conjugates of the arguments. One could conceivably extend Lemma 1 in this way by including conjugation, but this would not be compatible with the algebra of $\Gamma$-progressive functions, since $\Gamma$-progressive functions are not generally closed under complex conjugation. Under these conditions, then, Corollary 4 does not hold for general continuous activation functions.

If the preservation of conic support via the wavelets being $\Gamma$-progressive is not an issue, we can note further that when real-valued wavelets are being used, the Stone-Weierstrass theorem offers a way to understand uniform approximations of an INR architecture with continuous activation functions by INRs that satisfy the conditions of Lemma 1. For instance, ReLU activation functions are not polynomials, nor are they analytic, but uniform approximations by polynomials do exist over finite domains.

## B.2 Template Functions From Another Dimension

It is possible to consider cases where $\psi$ is defined as a map from a space of different dimension, say $\psi : \mathbb{R}^q \to \mathbb{C}$. If $q < d$, then one can construct a map $\tilde{\psi} : \mathbb{R}^d \to \mathbb{C}$ so that $\tilde{\psi}(x_1, \ldots, x_q, x_{q+1}, x_d) = \psi(x_1, \ldots, x_q)$. This, for instance, is the case with SIREN (Sitzmann et al., 2020) and the 1D variant of WIRE (Saragadam et al., 2023), where $\psi : \mathbb{R} \to \mathbb{R}$ is used for functions on higher-dimensional spaces. In this case, the Fourier transform of $\tilde{\psi}$ only exists in the sense of distributions. If $q > d$, then one can apply the results in this paper by treating the INR as a map from $\mathbb{R}^q \to \mathbb{C}$ followed by a restriction that "zeroes out" the excess coordinates by restricting the domain of $f_{\boldsymbol{\theta}}$ to the subspace of points $\mathbf{x}$ such that $\mathbf{x} = (x_1, \ldots, x_d, 0, \ldots, 0)$.

## C Proof of Corollary 4

We will show that the property of $f_{\boldsymbol{\theta}}$ being locally $\Gamma$-progressive at a point holds, as the result for being "globally" $\Gamma$-progressive follows.

Let $\mathbf{r}_0$ be given, and let $I \subseteq \{1, \ldots, F_1\}$ be the set of indices such that $\mathbf{W}_t\mathbf{r}_0 + \mathbf{b}_t \in \text{supp}(\psi)$. By Lemma 1, there exists an open neighborhood $U \ni \mathbf{r}_0$ such that for all $\mathbf{r} \in U$, $\mathbf{f}_{\boldsymbol{\theta}}(\mathbf{r})$ takes the form of a complex multivariate polynomial of $\{\psi(\mathbf{W}_i\mathbf{r} + \mathbf{b}_t)\}_{i \in I}$. Denoting this polynomial by $\mathcal{P}(\mathbf{r})$, we have that for any $\phi \in \mathcal{C}^\infty(U)$,

$$\phi \cdot \mathcal{P} = \phi \cdot f_{\boldsymbol{\theta}}.$$

Under the given assumptions, each of the terms $\{\psi(\mathbf{W}_i\mathbf{r} + \mathbf{b}_t)\}_{i \in I}$ is a $\Gamma$-progressive function. Since $\Gamma$-progressive functions constitute an algebra over $\mathbb{C}$, any polynomial of them will yield a $\Gamma$-progressive function. Letting $U$ be the "sufficiently small neighborhood of $\mathbf{r}_0$," this implies that $f_{\boldsymbol{\theta}}$ is locally $\Gamma$-progressive at $\mathbf{r}_0$, as desired.

Noting that $\Gamma$-progressive functions also constitute an algebra over $\mathbb{C}$ when $\Gamma$ is weakly conic, this result also applies in that case.

## D Wavelet Modulus Maxima Initialization in Two Dimensions

For 2D images, we approximate WMM location as edges of the image. Fig. 6 shows the flowchart for WMM-type initialization. We first start with the image to be represented and perform Canny edge detection on it. We then use the binary edge map as a proxy for WMM locations. We then use the edge locations as bias values for each neuron in the Gabor INR. The weights are initialized so that each neuron generates a radially symmetric complex Gabor wavelet, oriented such that the oscillatory direction is perpendicular to the detected edges.

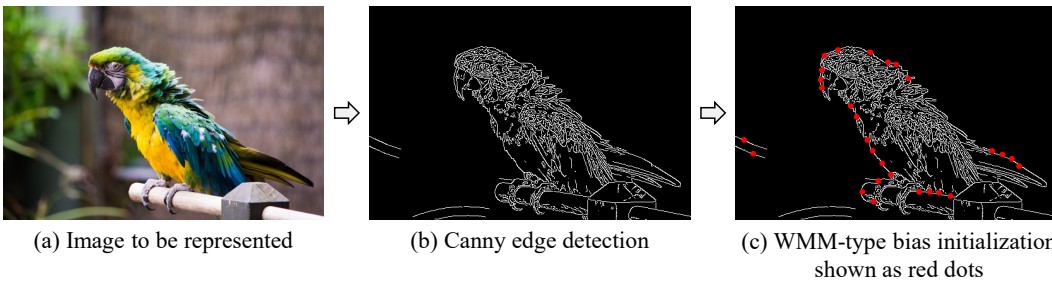

(a) Image to be represented  (b) Canny edge detection  (c) WMM-type bias initialization
shown as red dots

Figure 6: WMM initialization in two dimensions. (a) shows an image to be represented, while (b) shows the output of a Canny edge detector. (c) shows WMM-type bias initialization with the red dots showing some of locations used as bias values.

Table 1: Architectural descriptions for wavelet networks in Section 4

| Architecture | $\psi(x)$ | $F_1$ | $L$ | $F_2, \ldots, F_L$ | Parameters (wavelet+scaling=total) |
|---|---|---|---|---|---|
| Split, Complex, $L = 1$ | $\exp\left(-\frac{\pi^2 x^2}{6} + i2\pi x\right)$ | 2220 | 1 | n/a | $6661 + 257 = 6918$ |
| No scaling, Real | $\exp\left(-\frac{\pi^2 x^2}{6}\right)\sin(2\pi x)$ | 37 | 3 | 64 | $6731 + 0 = 6731$ |
| No scaling, Complex | $\exp\left(-\frac{\pi^2 x^2}{6} + i2\pi x\right)$ | 37 | 3 | 64 | $6731 + 0 = 6731$ |
| Split, Real | $\exp\left(-\frac{\pi^2 x^2}{6}\right)\sin(2\pi x)$ | 37 | 3 | 64 | $6731 + 257 = 6988$ |
| Split, Complex | $\exp\left(-\frac{\pi^2 x^2}{6} + i2\pi x\right)$ | 37 | 3 | 64 | $6731 + 257 = 6988$ |

## E  EXPERIMENTAL CONDITIONS

### E.1  ARCHITECTURAL DESCRIPTIONS FROM SECTION 4.3

We describe here both the architectures and training setups for the networks used in the experiments in Section 4.3. All scaling networks use gaussian template functions and no hidden layers beyond the one containing the template functions, *i.e.*, for all scaling networks,

$$\psi(x) = \exp\left(-\frac{\pi^2 x^2}{6}\right)$$
$$L = 1$$
$$F_1 = 128,$$

where the 128 template atoms are initialized to be uniformly spaced over the signal domain $[-2, 2]$.

Apart from the scaling networks shared by three of the five architectures used in Section 4.3, the wavelet networks considered have parameters listed in Table 1. All hidden layers use the activation function $\rho(z) = \sin(z)$, for $z \in \mathbb{C}$, where $\sin(z)$ is understood to be the analytic continuation of the real-valued sine to the complex plane. They are initialized according to the wavelet modulus maxima scheme described in Section 5.1, placing $K$ atoms at each maxima point, where $K = 1$ is chosen for architectures with hidden layers, and chosen in the $L = 1$ case to achieve a similar number of parameters to the other networks. The scales $\mathbf{W}^{(0)}$ are initialized uniformly at random in the interval $[1, K]$, so that the network with $L = 1$ has many scales of wavelets to fit a wide range of frequencies, while guaranteeing that the atoms are progressive at initialization.

All architectures are trained for a total of 4000 epochs using AMSgrad (Reddi et al., 2018) to minimize the mean-squared error between the real part of the INR and the target signal on $n = 512$ uniformly spaced points in the interval $[-2, 2]$. Split architectures are trained by first fitting the scaling network to the target signal for 2000 epochs, then fitting the scaling and wavelet networks simultaneously for 2000 additional epochs. Architectures that do not use a scaling network are trained for 4000 epochs.

### E.2 ARCHITECTURAL DESCRIPTIONS FROM SECTION 5.1

We describe here both the architectures and training setups for the networks used in the experiments in Section 5.1. For the examples fitting to the $1D$ test signal, the split complex Gabor architecture from Section 4.3 was used, as described in Appendix E.1.

For the examples fitting to test images, we use a WIRE INR (Saragadam et al., 2023) with $L = 3$ hidden layers for the wavelet network, with each $F_1, F_2, F_3 = 128$, and the same hidden nonlinearities as described in (Saragadam et al., 2023). For the scaling network, we use an INR with $L = 1$ and $F_1 = 128$, with a gaussian template function.

The network labeled "Gaussian" uses a 1D gaussian as a template function (see Appendix B.2), the network labeled "Real Gabor" uses the imaginary part of a complex Gabor wavelet as a template function, and the network labeled "Sine" uses a sinusoid as a template function (Sitzmann et al., 2020).

## F RESULTS ON KODAK IMAGE DATASET

To demonstrate some of the qualitative properties of INRs using complex template functions, we show here the individual representations of a subset of the images from the image fitting experiments in Section 5.1. Each image is pictured in Fig. 7 with its PSNR in dB relative to the ground truth. We exclude the representations by the real-valued Gabor network, due to similar observed behavior to the complex-valued version, as well as the representations using sinusoidal template functions, due to poor overall performance.

We first notice that the gaussian template function is outperformed by the Gabor wavelet for all pictured images, no matter what initialization scheme is used. The randomly initialized Gabor wavelet network occasionally outperforms the one using the WMM scheme, but this is by no more than $0.39$dB for the pictured images. The network initialized using the WMM scheme, on the other hand, has an advantage of up to $1.85$dB over the random initialization, indicating the strong advantage of using this scheme for images that are "cartoon-like," in the sense that they are smooth apart from isolated singularities such as edges.

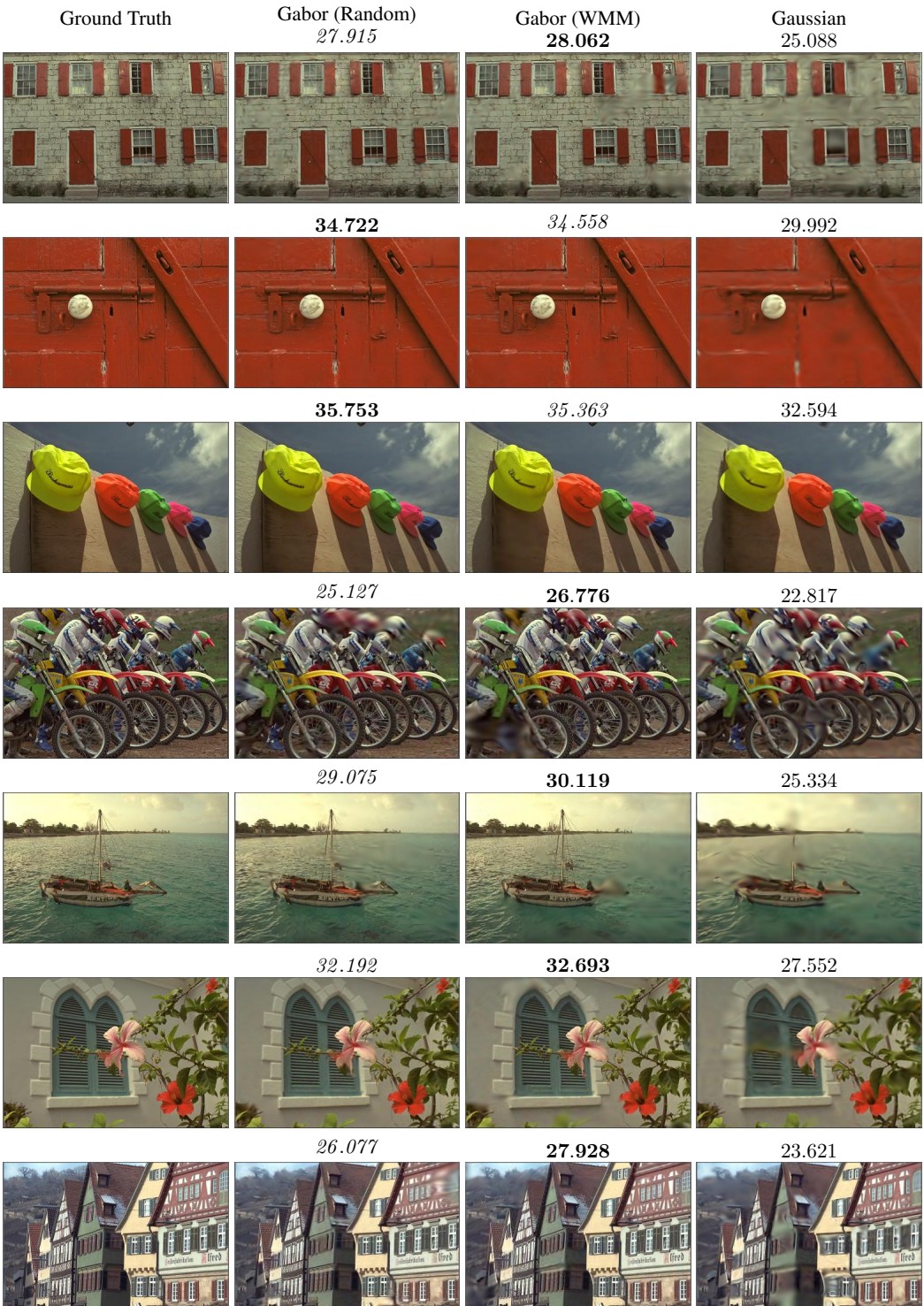

Figure 7: INRs fit to a subset of Kodak Lossless True Color Image Suite (kod, 1999). Images are labeled with their PSNR in dB relative to ground truth image, with the highest PSNR for a given image bolded and the second highest italicized.

