# OpenReview forum: "Implicit Neural Representations and the Algebra of Complex Wavelets"
_ICLR.cc/2024/Conference — ICLR 2024 poster_

### Official Review · Reviewer_ZxVa · 2023-10-13

**Soundness:** 4 excellent
**Presentation:** 3 good
**Contribution:** 3 good
**Rating:** 8
**Confidence:** 4

**Summary:**

This work studies the task of Implicit Neural Representations using neural networks with wavelet activation functions. Paper provides a theoretical framework for the approximation capabilities of such networks, and demonstrates how the theory can be helpful in designing the INR architectures. Experiments are provided on a few images.

**Strengths:**

The theory presented in the paper is a solid contribution for research and practice.

Paper is well written. Arguments are clear.

**Weaknesses:**

In my view, the weakness of this paper is its small set of experiments. I also think more details can be provided to interpret the experiments and to discuss the theory.

I think providing more details and more discussions would make the paper more approachable for a broader audience.

---------

Possible typo in the abstract: band-pass -> high pass. In my understanding, the method decouples the low-pass parts from the high-pass...

----------

I would have liked to see experiments on more images. I only saw three examples. Specifically, I'd be interested to see more examples of how the method works on more images, its errors, and its possible failure modes.

It may be useful if authors present cases where one might encounter troubles in training -- and cases where the learned representation may be flawed. The first image in Figure 6 seems to be more of a challenge from the training convergence perspective. Are there images where the reduction of the training loss would be even more challenging?

In the right column of Figure 6, the curves seem to still have a positive slope even towards the end of the horizontal axis. Is that correct? If yes, how would the curve proceed if training is continued further?

Authors can consider presenting the likes of figure 6 for earlier stages of training, e.g., for 5, 10, 50 epochs. How would those images look like? The convergence curve (right column in Figure 6) seems to be overly compact, so it is not easy to see the particulars for the early epochs of training. It seems that for the parrot picture, the WMM initialization lags behind the random initialization at the early epochs.

In Figure 6, the WMM result for the parrot has error at the left corners of the image. It may be useful if authors interpret those errors. Specifically, the patterns at the top left corner seem to appear in that general area of the image. Why does that magnitude of error only happen at that top left corner and not in that whole green area?

Overall, demonstrating results on more images would give a better understanding to a reader.

---------

I think the discussion in the context of wavelet literature could have been broader. For example, I did not see any discussions on the topic of Daubechies wavelets. Are Daubechies wavelets also progressive by the authors’ definition?

------

If authors think Shearlets may have any potential here, providing a discussion might be useful. Most of the approximation error for the pictures in Figure 6 appear to be at locations where the colors change in a small neighborhood. Could a shear matrix be potentially helpful in reducing the error because of its ability to extract anisotropic features?

------

A relevant prior work that authors may consider citing if they see fit:

-- Grattarola, D. and Vandergheynst, P., 2022. Generalised implicit neural representations. Advances in Neural Information Processing Systems, 35, pp.30446-30458.

-------

Using wavelets to study approximation properties of neural networks, and leveraging that to design the architecture of the network, is studied in the past, but I did not see a mention of that in the paper.

-- Shaham, U., Cloninger, A. and Coifman, R.R., 2018. Provable approximation properties for deep neural networks. Applied and Computational Harmonic Analysis, 44(3), pp.537-557.

**Questions:**

How can we interpret the result for the medical image in the bottom row of figure 6? To me, it seems that the error is more for the WMM case. Is that correct? If that is the case, the description in Appendix 5 may not be as accurate when it says “WMM based initialization has limited advantages …”. This would be a disadvantage for WMM and not an advantage?

Intuitively, how does the definition of progressive wavelets affect the approximation capability of a model? What would happen if we use a non-progressive wavelet to model the same images that authors present in Figures 1, 6, 7? We will lose the guarantees from the theorems, but how would that affect the learned representations? How would that empirically affect the error.

In Figure 6, how many parameters do the models have?

How could the theory and the method be used for image compression?

Please also see questions under weaknesses.

---

> ### Author Response · Authors · 2023-11-20
>
> Thank you for your thorough review of our work. We address some of the weaknesses below, as well as in the updated manuscript.

---

> > ### Author Response · Authors · 2023-11-20
> > **Discussion of Related Literature**
> >
> > 2. I think the discussion in the context of wavelet literature could have been broader. For example, I did not see any discussions on the topic of Daubechies wavelets. Are Daubechies wavelets also progressive?
> >
> > Daubechies wavelets, being real-valued in the usual case, are not progressive: we have added a comment on this in the updated paper. One can certainly convert any suitable wavelet to a progressive one by taking the Hilbert transform of the wavelet.
> >
> > 3. If the authors think Shearlets may have any potential here, providing a discussion might be useful. ... Could a shear matrix be potentially helpful in reducing the error because of its ability to extract anisotropic features?
> >
> > This is an interesting comment, and is indeed something that we have thought about. Empirically, architectures like WIRE have been observed by their authors to yield atoms in the first layer that resemble curvelets:
> >
> > Saragadam, Vishwanath, et al. "Wire: Wavelet implicit neural representations." Proceedings of the IEEE/CVF Conference on Computer Vision and Pattern Recognition. 2023.
> >
> > Seeing how the shear matrix of a shearlet transforms is used in conjunction with a parabolic scaling matrix to transform a "mother wavelet," INRs like the ones we study are capable of incorporating shearing transformations via the learning of the weights in the first layer of the network, without any explicit enforcing of the form of the matrix.
> >
> > However, the theoretical properties of systems of shearlets/curvelets are difficult to reconcile with INRs in the way we study them, as said properties are often asymptotic properties of a specifically designed system. Implemented naively, INRs can certainly use shearlets/curvelets in the first layer, but the higher powers of those atoms will probably not obey the relevant scaling rules that make said systems have nice theoretical properties. This is an interesting point for future research on the topic though, so we have made a note in the updated conclusion about this.
> >
> > 4. On some relevant works to cite from Grattarola, et. al., and Shaham, et. al.
> >
> > Indeed, thank you for pointing us to these references. We have included them in the survey section of the updated paper.

---

> > ### Author Response · Authors · 2023-11-20
> > **Possible Applications to Image Compression**
> >
> > 5. How could the theory and method be used for image compression?
> >
> > This is a good observation. Indeed, the WMM initialization scheme allows for an INR to express the singular information of an image better using a fixed number of neurons, since it doesn't rely on a random initialization to hopefully cover the singular parts of the image. We think that developing this into a more complete image compression method warrants a look into the training dynamics of INRs from a harmonic analysis perspective, which is a great direction for future work.
> >
> > And of course, many methods for meta-learning from datasets, pruning, and so on can be applied to generically compress the set of weights in a neural network, which applies to INRs as well.

---

> ### Author Response · Authors · 2023-11-20
> **Empirical Parts of the Paper**
>
> 1. Need for more experiments. How many parameters do the models have?
>
> To address these remarks, we have added new comparative experiments to Section 4.3 showing the interaction between the use of complex wavelets as template functions and the proposed split INR architecture. We have also added experiments to Section 5 showing the advantage of the wavelet modulus maxima initialization scheme for a dataset of images. This resulted in the figure showing the training behavior of the networks being removed to make space for more substantive results.
>
> We have added an appendix describing the models in greater detail, including the depths and widths of the networks.

---

> ### Comment · Reviewer_ZxVa · 2023-11-22
>
> I thank the authors for their clear and detailed response. The revised paper appears to me as a good contribution and my recommendation is acceptance.
>
> Below are some comments that authors might find useful. I'm not asking the authors to provide a response as the rebuttal period is closing soon.
>
> Comment 1: In the previous version of the paper, I found the error plots of the approximated images insightful, especially the one for the parrot image, but it seems that they are excluded in the revision. For the images in Figure 7, I can recognize some of the regions that are the source of error. Having the error plots would have made it easier to identify and compare those regions. I respect that authors might not want to include the error plots for some reason.
>
> Comment 2: Again in Figure 7, the blurred parts in the reconstructed images vary across the columns (Gabor-random vs Gabor-WMM vs Gaussian). Perhaps this deserves further discussion in the appendix. What is the sensitivity of the resulting images.
>
> Comment 3: For the images in Figure 7, if one compresses those images with a compression method such as JPEG, which parts of the images would be blurred, and how would the resulting blurred regions be different than the blurred regions obtained by authors' approximation method.

---

### Official Review · Reviewer_rBV3 · 2023-10-30

**Soundness:** 2 fair
**Presentation:** 2 fair
**Contribution:** 3 good
**Rating:** 6
**Confidence:** 4

**Summary:**

The article studies implicit neural representation models which typically uses sinusoidal activation functions and wavelet-based activation functions, in order to represent signals. This work develops theoretical understanding of such architectures and also a practical split architecture to capture singularities in target functions.

**Strengths:**

The theoretical analysis seems to be novel. By using Fourier analysis, theorem 1 gives an ide a about the effective time-frequency support of implicit neural representations. Numerical results such as in Fig 4 further support the theory.

**Weaknesses:**

The overall clarity of the presentation still needs to be improved to make results accurate. See questions below.

**Questions:**

-	Your definition of the model in eq 1 seems to have some issue to me. Psi is a function of R^d -> C but somehow W^0 r and b^0 is in R^{F_1 x d}. Thus I do not understand the model.
-	The notation inf Theorem 1 is not clear. What is the definition of the Fourier transform for functions in C_0^inf (U) in eq 2? What does this stand for C_0^inf (U)? What is the product_t from t=1 to t=F_1? Does hat beta_l depends on W^{l} and b^{l}? Is * a convolution, over which domain? As a consequence, the argument in Section 3.2 regarding the support of the product_t term in eq. 2 is not clear. Is W^T a typo of W_l^T?
-	In Section 4.3, what does it mean to model a signal as a sum of a linear scaling INR and a nonlinear INR ?

---

> ### Author Response · Authors · 2023-11-20
>
> Thank you for your comments on our paper. We hope that the updates to the paper corresponding to our replies to your questions have improved the presentation of the work.

---

> > ### Author Response · Authors · 2023-11-20
> > **Resolving some Notational and Technical Concerns**
> >
> > 1. $\psi$ is a function from $R^d$ to $C$, but somehow $W^0r$ and $b^0$ are in $R^{F_1\times d}$.
> >
> > Good point: the function $\psi$ is understood to act "row-wise" on the matrix in $R^{F_1\times d}$, so that it extends to a function $\psi:R^{F_1\times d}\to C^{F_1}$. We have added a comment to this effect in the updated paper.
> >
> > 2. The notation in Theorem 1 (now Lemma 1) is not clear. What is the definition of the Fourier transform for functions in $C_0^\infty(U)$? What does this stand for?
> >
> > $C_0^\infty(U)$ is standard notation to denote the set of infinitely differentiable functions whose support is compact and contained in $U$. We have added a note clarifying this definition in the paper.
> >
> > 3. What is the product from $t=1$ to $t=F_1$?
> >
> > The large asterisk denotes a convolution of the terms indexed by $t=1,...,F_1$. We have added a note clarifying this notation to the paper.
> >
> > 4. Does $\hat{\beta}_l$ depend on $W^l$ and $b^l$?
> >
> > We see the confusion here; indeed, we should not have used the lowercase letter $\ell$ here in more than one way. We have changed the index notation to remove this ambiguity. Thank you for pointing this out.
> >
> > 5. Is $*$ a convolution, over which domain?
> >
> > Yes, it is a convolution in the Fourier domain.
> >
> > 6. The argument in Section 3.2 regarding the support of the product term in Eq. 2 is not clear.
> >
> > Based on comments of other reviewers, we have removed this section from the updated paper in order to make room for other additions to the paper. Thank you for pointing out the potential typo, though!
> >
> > 7. In Section 4.3, what does it mean to model a signal as a sum of a linear scaling INR and a nonlinear INR?
> >
> > We have updated Section 4.3 with new comparative experiments to study the proposed architecture, along with a more precise explanation of what we mean by this. Thank you for suggesting this clarification, we feel that it has improved the presentation of this section of the paper.

---

> > > ### Comment · Reviewer_rBV3 · 2023-11-21
> > > **accept**
> > >
> > > Dear authors,
> > > Thanks for the clarification and your revised version. I am glad to accept the article as the algebra of the progressive wavelets is very nice to construct the INR model. I think this is the main novelty rather than your theorem 1 which seems to be a basic application of the convolution theorem. I did not understand your proof in Appendix A ... please make it more clear as much as you can.

---

### Official Review · Reviewer_ewcV · 2023-10-31

**Soundness:** 4 excellent
**Presentation:** 3 good
**Contribution:** 3 good
**Rating:** 6
**Confidence:** 3

**Summary:**

The study conducts a time-frequency analysis of INRs by leveraging polynomial approximations to delve into the behaviors of MLPs beyond the first layer. By decomposing a signal into its low-pass and high-pass segments using two INRs, motivated from the scaling and wavelet functions of the wavelet transform. This approach bridges the structure of complex wavelets and the application of INRs, which is a novel concept.

**Strengths:**

This paper presents a novel perspective on behaviors of INR models, including the decomposition of low and band-pass approximations, along with specific initialization methods, and enhances both the depth and practical relevance of the study.

**Weaknesses:**

Although the performance of the proposed method was supported by several tests and analysis in the paper, it is suggested to include some practical applications such as regression tasks on images or other high-dimensional signals to justify its practicability.

**Questions:**

The reviewer is curious if the proposed method and its theoretic analysis/intuition is valid, when the method is applied to practical applications.

---

> ### Author Response · Authors · 2023-11-20
> **New Experiments Added**
>
> Thank you for your comments on our work. You are indeed in agreement with the other reviewers on the need for more numerical experiments to go with the paper, which we have addressed by updating Sections 4 and 5.
>
> We would like to remark that the main point of this paper is not strictly methodological. Rather, the suggestion of using a split architecture for INRs that use progressive wavelets as template functions is something that follows from our analysis of such networks, as determined by the Theorem (now known as Lemma 1) and the following discussion.
>
> With this in mind, we still see the need for more practical demonstrations to show the validity of the obtained insights. In Section 4.3, you can see a comparative study between the use of real and complex wavelets, the use of nonlinear hidden layers in INRs, and using scaling networks or not (i.e., the proposed split architecture).
>
> We have also added new experiments on a dataset of images showing the advantage of initializing a complex Gabor INR using the wavelet modulus maxima over random initialization, as well as over other choices of template function.

---

### Official Review · Reviewer_3UQT · 2023-11-03

**Soundness:** 3 good
**Presentation:** 3 good
**Contribution:** 3 good
**Rating:** 6
**Confidence:** 3

**Summary:**

The authors use Fourier analysis to study the properties and expressive power of implicit neural representations (INR). In particular, they analyze multi-layer INRs whose first-layer activations (which they call the template function) map from Euclidean space into the complex numbers and whose activations on all other layers are polynomials.

The authors show that when the template function is essentially a band-pass filter, the class of INRs they study also act essentially as band-pass filters. Based on this observation, the authors propose to model signals as a sum of two INRs instead, one that acts as a low-pass filter and one that acts as a high-pass filter.

The authors also suggest initializing INRs with wavelet template functions placed on estimate singular (i.e., non-smooth) points in the domain of the signal (e.g., edges in an image). They show that this careful initialization results in significantly improved performance compared to random initialization.

**Strengths:**

Analyzing the expressivity of multi-layer INRs, not just one-hidden-layer ones, is a very relevant problem; hence, any progress in the area is nice. I found the low-pass-high-pass decomposition idea and the suggestion to initialize the template function's biases to the signal's singular points interesting.

**Weaknesses:**

As someone whose primary area of research or background is not in classical signal processing, I found the paper quite confusing. In particular, the authors provide little to no interpretation of their results and observations. This led me to feel that I was constantly expected to be able to interpret and understand them, which I failed to do in many cases. Moreover, it is unclear what mathematical level the authors expect of the reader. They state some quite elementary results rigorously while handwaving others (e.g., effective support, Minkowski sums). Finally, several technical signal processing terms (such as atoms, band-pass filters, and WIRE) are undefined, which makes reading the paper quite challenging.

For example, the authors showcase Thm 1 as one of their main results. However, Eq (2) is just the Fourier transform of a slightly rearranged definition of the INR architecture they are studying. On the other hand, some of the assumptions and both conclusions are unclear:
- Why is restricting our focus to polynomial activations beyond the first layer interesting to study?
- Why is the Fourier transform stated by considering multiplication by a smooth function $\phi$?
- How does the fact that the INR is composed of the sum of the template function's "integer harmonics" illustrate the expressivity of the INR?
- "Second, the support of these scaled and shifted atoms is preserved so that the output at a given coordinate $r$ is dependent only upon the atoms in the first layer whose support contains $r$." - How is this not a trivial statement? What is the relevance of this observation?

Similarly, what is the purpose of section 3.2? While I follow and agree with the argument, the whole discussion is informal, so it technically contains no actual results. More importantly, though, it's unclear what the insight is.

However, Section 4 is perhaps the most confusing part of the paper. In the first paragraph, the authors state that in this section, they "consider the advantages of using complex wavelets, or more precisely progressive wavelets." This sentence has two issues: first, it shows that the paper title is a misnomer because the authors actually only consider the algebra of progressive wavelets. Second, they don't demonstrate any advantages of using progressive wavelets. Perhaps most confusingly, even though the concept is featured in the paper title, the authors never actually make use of the fact that progressive wavelets form an algebra.

Essentially, the authors spend the first two subsections defining a multivariate notion of a band-pass filter using weakly conic sets. Then, in the third section, the authors claim in Section 4.3: "Based on this property of INRs preserving the band-pass properties of progressive template functions, it is well-motivated to approximate functions using a sum of two INRs: one to handle the low-pass components using a scaling function, and the other to handle the high-pass components using a wavelet." I do not follow this argument. I'm not saying it is not a good idea, but I do not see why it is well-motivated based on the multivariate notion of band-pass filters to decompose signals into a sum of a high-pass INR and a low-pass INR.

Finally, the experimental section of the work is severely lacking. The authors only conducted one ablation study for their suggested initialization technique vs. random initialization by reconstructing three different images. Hence, it is unclear how the suggested signal decomposition into two INRs or the initialization technique helps with the usual tasks INRs solve in practice compared to other methods.

**Questions:**

n/a

---

> ### Author Response · Authors · 2023-11-20
>
> Thank you for your detailed review. Your comments in conjunction with the other reviewers have led us to restructure some aspects of the paper, as well as add in new experiments. We address your comments below.

---

> > ### Author Response · Authors · 2023-11-20
> > **Discussion of Main Result and Interpretation**
> >
> > 2. Eq (2) is just the Fourier transform of a slightly rearranged definition of the INR architecture they are studying.
> >
> > This is a fair point, and perhaps we did not emphasize the correct aspects of the paper. The point of the work is not Theorem 1 on its own, but rather to see how it interacts with different choices of the template function, in particular template functions that constitute an algebra under multiplication. To shift the emphasis, we have "downgraded" Theorem 1 to a Lemma, and provided a comment remarking that it is really just an application of the convolution theorem.
> >
> > 3. How does the fact that the INR is composed of the sum of the template function's "integer harmonics" illustrate the expressivity of the INR?
> >
> > This statement gives a rough bound on the type of function that can be expressed by the INR. That is to say, it dictates what functions can and cannot be represented by an INR. For instance, if the template functions in the first layer all have Fourier support in some cone (that is, they are $\Gamma$-progressive for some conic set $\Gamma$), then the INR will be unable to generate frequencies outside of $\Gamma$. Depending on what an INR user is looking for, this could be viewed as a helpful or harmful constraint, but one worth knowing about either way.
> >
> > 4. "The support of these scaled and shifted atoms is preserved so that the output at a given coordinate is dependent only upon the atoms in the first layer whose support contains..." -- How is this not a trivial statement? What is the relevance of this observation?
> >
> > Indeed, this is a simple property of the considered INRs -- that statement was meant merely to state the last part of Theorem 1 (now Lemma 1) in words. The observation becomes relevant when we consider the later sections on the use of $\Gamma$-progressive functions, as it is unreasonable to expect a multidimensional function (such as an image) to be globally approximated by a $\Gamma$-progressive function for some nontrivial conic set $\Gamma$, but it is more reasonable to expect this to hold locally in the image. See the above example of the unit disc.
> >
> > 5. What is the purpose of Section 3.2?
> >
> > This is a fair critique -- based on your comments, as well as those of other reviewers asking for some additions to the paper, we have removed Section 3.2 from the updated manuscript to make room for new material.

---

> > ### Author Response · Authors · 2023-11-20
> > **Discussion of Technical Details**
> >
> > 6. Why is restricting our focus to polynomial activations beyond the first layer interesting to study?
> >
> > Polynomial activations, although not usually used directly in neural network architectures, are reasonable ways to approach the harmonic analysis of more general neural network architectures. We show how polynomial activations can be extended to analytic (holomorphic) activations and beyond in Appendix B.1. For the case of real-valued networks, for instance, any continuous neural network can be uniformly approximated by polynomials over a compact domain by the Stone-Weierstrass Theorem, a fact leveraged in the study of ReLU networks in the paper
> >
> > Mehmeti-Göpel, Christian HX Ali, David Hartmann, and Michael Wand. "Ringing ReLUs: Harmonic distortion analysis of nonlinear feedforward networks." International Conference on Learning Representations. 2020.
> >
> > Of course, complex numbers complicate the choice of activation functions a bit in this case. For instance, the modulus function is not holomorphic, and thus can't be approximated by polynomials without using conjugation, and progressive functions are not closed under conjugation in general, so the interplay between Theorem 1 (now Lemma 1) and the algebra of progressive wavelets does not apply here. We have remarked upon this in the updated paper and in the appendix.
> >
> > 7. Why is the Fourier transform stated by considering multiplication by a smooth function?
> >
> > This is done for two reasons: the first is technical, the second is a matter of interpretation/emphasis.
> >
> > The technical reason is to handle Fourier transforms of functions that are only locally integrable. ReLU functions, for instance, are only integrable locally, not globally, so their Fourier transform only exists in the sense of tempered distributions (i.e., via duality with the space of rapidly decaying smooth functions). Multiplying by a compactly supported smooth function assuages these issues, and can be used to reconstruct the Fourier transform of the whole signal if needed.
> >
> > The interpretive reason is to emphasize the locality of the INR architecture. In the time-frequency analysis perspective that we take, it is important to understand wavelets and their powers as being localized in space. This has important interactions with the notion of a locally $\Gamma$-progressive function for some conic set $\Gamma$, since the singularities in a function might only have common direction locally. Take the indicator function of the unit disc in $R^2$, for instance. The singularities of the disc are located along the edges of the disc and point in the normal direction to the boundary. Because of this, the Fourier transform of this function has slow decay in all directions. However, if you localize to a small neighborhood of a point on the boundary of the disc via multiplication by a suitable smooth function, the Fourier transform will only exhibit slow decay in a small cone around the normal direction. This is where the notion of a conically progressive function becomes useful, as you can guarantee that all generated frequencies are in line with the direction of a desired singularity.

---

> ### Author Response · Authors · 2023-11-20
> **Terminology and Writing**
>
> 1. [For those whose ] background is not in classical signal processing, ... the paper [is] quite confusing. The authors provide little to no interpretation of their results and observations. It is unclear what mathematical level the authors expect of the reader. Several signal processing terms are undefined, which makes reading the paper quite challenging.
>
> Thank you for pointing this out. While certain terminology is quite standard to the intended audience of our paper (broadly speaking, those that are well-acquainted with Fourier analysis and its applications), we would still like to avoid using excessive jargon. The updated paper has improved explanations behind the motivations for certain results and proposed methods. Additionally, based on your comments and comments from other reviewers, we have removed Section 3.2 to make more space for new results and exposition, which seemed to be a problematic section anyway.

---

> ### Author Response · Authors · 2023-11-20
> **Motivation for Proposed Methods and Experiments**
>
> 8. The authors state in [Section 4] that they "consider the advantages of using complex wavelets, or more precisely progressive wavelets." ... Second, they don't demonstrate any advantages of using progressive wavelets. Most confusingly, ..., the authors never actually make use of the fact that progressive wavelets form an algebra.
>
> Thank you for this comment: it has revealed some aspects of the paper that we should have made more clear to begin with. In conjunction with added comparative experiments, we have updated Section 4.3 to empirically demonstrate the advantages of using progressive wavelets over their real-valued counterparts.
>
> The titling of the paper with the term "complex wavelet" rather than "progressive wavelet" was a stylistic choice, as "progressive wavelet" is not a universally-used term in the literature.  Many signal processing researchers will informally use the term "complex wavelet" to refer to what we call "progressive wavelets" in this paper, although it is certainly true that not all complex-valued wavelets are necessarily progressive. If the area chair and other reviewers also think that a change in title is appropriate, we are willing to do so.
>
> We do make use of the fact that progressive wavelets form an algebra: Corollary 4 is a combination of Theorem 1 (now Lemma 1) and the algebraic closure of the space of progressive functions.
>
> 9. The authors claim in Section 4.3: "Based on this property on INRS... it is well-motivated to approximate functions using a sum of INRs..."
>
> Indeed, we could have made this more clear from the start. We have updated Section 4.3 to address this in two ways. We have clarified why it is well-motivated to split the signal representation in this way by explaining that INRs using progressive wavelets are unable to easily generate low-frequency signal content. Additionally, we have noted the potential advantages of separating the low-pass and high-pass parts of the signal from a signal processing perspective. In line with this discussion, the added experiments show empirically the advantage of the split architecture with complex wavelets over alternative design choices.
>
> 10. Finally, the experimental section of the work is severely lacking. ... it is unclear how the suggested signal decomposition into two INRs or the initialization technique helps with the usual tasks INRs solve in practice compared to the other methods.
>
> Thank you for this comment. You are in agreement with the other reviewers that this work needs more empirical demonstrations. We have updated Section 4.3 with a comparative study of using the split architecture vs INRs that do not have a "scaling network" attached, as well as a discussion of why the split architecture with complex wavelets may outperform the other approaches.
>
> We have also updated Section 5 with an experiment testing the initialization scheme on a larger dataset of images, comparing to other template functions from the literature, and showing how the initialization is helpful in denoising tasks.

---

> > ### Comment · Reviewer_3UQT · 2023-11-21
> > **Response to the authors**
> >
> > I thank the authors for their elaborate response. I have skimmed through the updated version of the paper, spending more time on sections that I found unclear in the previous version. I am satisfied with the updated version, and am now happy to recommend its acceptance; I have increased my score to reflect this.
> >
> > In my opinion, the authors have made their paper a great deal more accessible and also more interesting to the ICLR community by including more explanations and experiments on Kodak.
> >
> > As a final point, I find that the authors' explanation for the relevance of Lemma 1 in the rebuttal is still clearer than the one in the paper. Hence, I think the authors could further strengthen the paper by incorporating more of the points they mention in the rebuttal into the camera-ready version.

---

> > > ### Author Response · Authors · 2023-11-21
> > > **Thank you for your updated review**
> > >
> > > Thank you for considering our revisions and updating your review -- we are glad to hear that you found the paper to be improved based on your comments and those of the other reviewers. We will be sure to incorporate more of the discussion around Lemma 1 from the review process as we finalize the paper.

---

### Official Review · Reviewer_ksCd · 2023-11-06

**Soundness:** 4 excellent
**Presentation:** 4 excellent
**Contribution:** 2 fair
**Rating:** 8
**Confidence:** 3

**Summary:**

This paper studies a type of Implicit Neural Representation based on a wavelet nonlinearity.
An implicit neural representation is a parameterization of a (usually) scalar function of a (low-dimensional variable), e.g. a function f: R^d->R using a neural network. In this setting, an image is a function R^2->R and the input to the function are pixel locations. This setup requires some reflection about the role of in particular the first non-linearity. Using ReLU will lead to lots of ramps, requiring their combination in intricate ways to represent usual real-world signals.
Here the use of wavelets as first non-linearity is analyzed. The use of wavelets comes out of a line of research leading via sinusoidal components (e.g. SIREN) and presents itself as a clear follow-up. The setting studied here is a first layer with wavelet nonlinearity followed by several layers of pointwise mixing layers with polynomial nonlinearities.

In this context, using the Fourier convolutional theorem, the functions expressible by this architecture are characterized in Fourier space as essentially a collection of higher-order self-convolutions of affine-transformed versions of the base wavelets.

Introducing the notion of "progressive wavelet" which have their Fourier support on a cone (or a "weak cone" that is closed for factors >= 1), it is shown that the wavelet self-convolutions above never leave their cone, leading to a neat characterization of what functions can be matched using a particular wavelet.

To accommodate low-pass signal, a split of the INR is proposed into the sum of two INRs, one using a wavelet, the other using a low-pass filter. This decomposition is shown to be beneficial in fitting a signal.

Further, wavelet modulus maxima points are proposed to initialize the representation. It is shown that for certain signals the use of WMM points as initialization leads to a better fit.

**Strengths:**

The paper is very clearly written and presents its contributions is a highly succinct way. It is a pleasure to read - in fact it reads a bit like an advanced chapter of a wavelet applications textbook. The illustrations contribute to the ease of understanding.

The paper gives a concise characterization of the function space spanned by the wavelet nonlinearity followed by layers of pointwise polynomial mixing. From this characterization it clearly identifies, using progressive wavelets, that a split into low-pass and wavelets is highly useful.

In short, it leads to a complete understanding of the particular setting introduced. The hope is that this understanding can extend to adjacent settings.

**Weaknesses:**

The provable statements in the paper are not in any way non-obvious. Theorem 1 is a direct consequence of the Fourier convolution theorem.

The setting in which these proofs work are highly impoverished with respect to the setting of actual interest, which is that of non-polynomial nonlinearities, such as the ReLU, for the pointwise mixing layers. The shifting of frequencies along a cone does not hold for these, and complicated ringing processes emerge that can also define sharp boundaries by the rectifier suppressing negative values. (The effects of ReLU wrt wavelets are partially analyzed e.g. here https://arxiv.org/abs/1810.12136 but it looks intractable to integrate this in the current analysis).

Given the nature of the expressed function space, what is an obvious advantage of this particular implicit neural representation over a sparse continuous wavelet transform with a sufficiently expressive bank of filters (e.g. some base filters and some of their polynomial powers)? In particular, if one needs to use wavelet modulus maxima to initialize the representation. Could these advantages be concisely stated? (e.g. I can imagine that potentially the representation requires fewer parameters)

In general, could some numerical comparisons be done to place the analyzed method within context of other INRs? We are not looking for state of the art here, but to have an idea of whether the proposed setup is close or far from that. If it is close, this can also partially alleviate the expressivity concern mentioned above.

**Questions:**

The term "progressive wavelet" was new to me. However, as far as I can tell, its definition fits exactly what I know as "analytic wavelets". Could the authors confirm this is the same and in that case justify the use of a new term, or explain in what way these concepts differ?

There are two more direct questions listed in the weaknesses section.

Regarding the low-pass + progressive wavelet decomposition for images: Is it possible to show the smooth parts and the wavelet parts? A conjecture for this, if both INRs are allowed to output in RGB space, is that the smooth parts contain much more color than the wavelet parts. Would be interesting to see if that is the case for certain natural images.

---

> ### Author Response · Authors · 2023-11-20
>
> Thank you for your detailed review of our paper; your comments have directly led to an improved presentation of the work. We address your comments and questions below, first addressing those from the weaknesses section, and then those from the questions section.

---

> > ### Author Response · Authors · 2023-11-20
> > **Concerns about Theoretical Aspects**
> >
> > 1. The provable statements in the paper are not in any way non-obvious. Theorem 1 is a direct consequence of the Fourier convolution theorem.
> >
> > This is a fair point; indeed, all that Theorem 1 expresses is the Fourier transform of a polynomial of functions in terms of convolutions in the Fourier domain. The value in the particular expression comes from the later interaction with the notion of a progressive template function, such as a wavelet or complex exponential. We have changed the paper by "downgrading" Theorem 1 to a Lemma, with a comment acknowledging the simplicity of the result.
> >
> > 2. The setting in which these proofs work is highly impoverished with respect to the setting of actual interest, which is that of non-polynomial nonlinearities, such as the ReLU. The shifting of frequencies along a cone does not hold for these, and complicated ringing processes emerge...
> >
> > We would first like to note that analytic activation functions, such as the sinusoidal activation used in SIREN, was shown in
> >
> > Sitzmann, Vincent, et al. "Implicit neural representations with periodic activation functions." Advances in neural information processing systems 33 (2020): 7462-7473.
> >
> > to in fact outperform ReLU activation functions. Although the Theorem (now, Lemma 1) was stated for finite-degree polynomials, there is a corresponding result for general analytic activation functions in Appendix B.1, which is referenced after the statement of Lemma 1. Moreover, for real-valued networks with ReLUs, the paper
> >
> > Mehmeti-Göpel, Christian HX Ali, David Hartmann, and Michael Wand. "Ringing ReLUs: Harmonic distortion analysis of nonlinear feedforward networks." International Conference on Learning Representations. 2020.
> >
> > studies the Fourier properties of ReLU networks by approximating the nonlinearities by polynomials, via the Stone-Weierstrass theorem. In our case, the nonlinearities need to act on complex numbers, so that polynomials (without conjugation) can only approximate holomorphic activation functions, excluding options such as the modulus. It is a reasonable critique, so we have added a comment addressing this in the relevant appendix of the paper.
> >
> > In the linked paper
> >
> > Mallat, Stéphane, Sixin Zhang, and Gaspar Rochette. "Phase harmonic correlations and convolutional neural networks." Information and Inference: A Journal of the IMA 9.3 (2020): 721-747.
> >
> > they do study relationships between rectifiers (ReLUs) and complex wavelet coefficients, but this is done in a way so that the rectifier is only applied to the real part of a complex signal. The ultimate goal of their approach is to understand the phase of the wavelet transforms of real-valued signals after rectifiers are applied. This,as you note, doesn't quite fit with the type of statements made in our work, as it is concerned with activation functions applied to complex-valued signals directly.

---

> > ### Author Response · Authors · 2023-11-20
> > **Comparative Experiments**
> >
> > 3. What is an obvious advantage of this particular implicit neural representation over a sparse continuous wavelet transform with a sufficiently expressive bank of filters (e.g., some base filters and some of their polynomial powers)? In particular, if one needs to use wavelet modulus maxima to initialize the representation. Could these advantages be concisely stated?
> >
> > This is a very good point: we have added in more comparative experiments to Section 4 of the paper, including a comparison to a "sparse continuous wavelet transform," which we treat as an INR with no hidden layers beyond the template functions. We designed the sparse CWT INR to have a similar number of parameters to the architecture that we suggest, and observe that it attains a higher MSE on the test signal.
> >
> > Beyond an empirical demonstration, there are a few reasons why an INR with polynomial/analytic activations may be preferred over a bank of filters as described. By coupling the wavelet atoms and their powers via the hidden layers, an INR can generate many coefficients that are correlated to each other in order to resolve a singularity. While a standard wavelet transform, or one using powers of wavelets as well, can also do this by careful selection of the scales and abscissa (this is precisely the statement of Lemma 1!), decoupling these coefficients means that each one needs to be considered independently. Of course, the coupling across scales reduces the degrees of freedom of an INR with hidden layers compared to a large bank of wavelets, but the way in which this coupling is done perhaps introduces an implicit bias that makes learning easier.
> >
> > 4. Could some numerical comparisons be done to place the analyzed method within context of other INRs?
> >
> > Yes, we have added in more numerical experiments with a variety of architectural choices. See the updated Section 4.3 for more comparisons of using scaling networks vs not (that is, testing the "split" architecture) in both the regimes of real and complex wavelets. This has led to a new observation that makes sense in hindsight, but was not obvious from the results of the paper: when using a split architecture, complex wavelets perform better, but when not using a network to approximate the signal with scaling functions, it is preferable to use real wavelets. This is due to the fact that powers of progressive wavelets do not generate low-frequency signal content, while power of real-valued wavelets are capable of doing so.
> >
> > We have also added in new experiments demonstrating the proposed approach on a dataset of images, also comparing to other choices of nonlinearities that have been used in INRs.
> >
> > 5. Is it possible to show the smooth parts and the wavelet parts in the low-pass+progressive wavelet decomposition?
> >
> > We have pictured this in Figure 4 of the paper for the 1D test signal. As expected, the scaling network yields a smooth signal, while the wavelet network is close to zero apart from the singularities.
> > In addition to showing how the low-pass and high-pass parts are captured respectively by the scaling and wavelet networks, we also show in the bottom row how the nonlinearities resolve the high-pass features from a set of relatively smooth template functions via the hidden nonlinearities.

---

> ### Author Response · Authors · 2023-11-20
> **Terminology Question**
>
> 6. Does the term "progressive wavelet" mean the same thing as "analytic wavelet?"
>
> Yes indeed -- we chose to use the term "progressive wavelet" to avoid confusion with the "analytic" activation functions of the INR. The term progressive wavelet is used in this way in the literature, such as by
>
> Grossmann, Alexandre, Richard Kronland-Martinet, and J. Morlet. "Reading and understanding continuous wavelet transforms." Wavelets: Time-Frequency Methods and Phase Space Proceedings of the International Conference, Marseille, France, December 14–18, 1987. Berlin, Heidelberg: Springer Berlin Heidelberg, 1990.

---

### Author Response · Authors · 2023-11-20
**Paper Updated to Incorporate Reviewer Feedback**

Dear Reviewers,

Thank you for the time and effort you put in to make such a thorough review of our manuscript. We have updated the manuscript, as well as responded to your comments below. In summary, the updated manuscript clarifies notational, technical, and expositional concerns of the reviewers, as well as adds in new experiments to further demonstrate the consequences of the theoretical results and proposed architectures, as well as to place the proposed methods in broader context of the literature on INRs.

We are happy to answer any further questions you may have.

All the best,
Authors

---

### Meta-Review · Area_Chair_We1f · 2023-12-06

**Metareview:**

All reviewers agree that this paper discusses an important perspective on implicit neural learning with wavelet activation functions and provides solid theoretical results for this problem. Although some reviewers have raised concerns regarding empirical evaluations and some theretical assumptions or explanations for those, it appears that the authors could properly address these points and modify the paper according to the comments by the reviewers. Overall, I recommend the decision of acceptance (poster) for this paper.

**Justification For Why Not Higher Score:**

Although the authors adequately address the weaknesses of the paper raised by the reviewers, the theoretical results seem to be open to be further modification in some points such as theoretical assumptions or explanations in addition to empirical justification.

**Justification For Why Not Lower Score:**

All reviewers agree that this paper discusses an important perspective on implicit neural learning with wavelet activation functions and provides solid theoretical results for this problem. I think therefore the paper should be published in this venue.

---

### Decision · Program_Chairs · 2024-01-16

Accept (poster)